# Physical Stability of Oil-In-Water Emulsion Stabilized by Gelatin from Saithe *(Pollachius virens)* Skin

**DOI:** 10.3390/foods9111718

**Published:** 2020-11-23

**Authors:** Pauline Henriet, Flemming Jessen, Mar Vall-llosera, Rodolphe Marie, Mastaneh Jahromi, Mohammad Amin Mohammadifar, Hanne Lilian Stampe-Villadsen, Heidi Olander Petersen, Jens J. Sloth, Karin Loft Eybye, Greta Jakobsen, Federico Casanova

**Affiliations:** 1Agrocampus Ouest, UMR 1253, F-35042 Rennes, France; pauline.henriet@agrocampus-ouest.fr; 2Research Group for Food Production Engineering, National Food Institute, Technical University of Denmark, Søltofts Plads, 2800 Kongens Lyngby, Denmark; fjes@food.dtu.dk (F.J.); marvallju@gmail.com (M.V.-l.); mastanehjahromi@gmail.com (M.J.); moamo@food.dtu.dk (M.A.M.); hjac@food.dtu.dk (H.L.S.-V.); hope@food.dtu.dk (H.O.P.); 3Department of Health Technology, Technical University of Denmark, Ørsted Plads, 2800 Kongens Lyngby, Denmark; rcwm@dtu.dk; 4Research Group for Nano-Bio Science, National Food Institute, Technical University of Denmark, Kemitorvet, 2800 Kongens Lyngby, Denmark; jjsl@food.dtu.dk; 5Technological Institute, Kongsvang Alle 29, DK-8000 Aarhus C, Denmark; klt@teknologisk.dk; 6Danish Fish Protein, Adelvej 11, Hoejmark, DK-6940 Lem St, Denmark; gj@danishfishprotein.dk

**Keywords:** saithe (*Pollachius virens*) skin, gelatin, O/W emulsion, physical stability

## Abstract

The objective of the present study was to investigate the physical stability of an oil-in-water (O/W) emulsion stabilized with gelatin from saithe (*Pollachius virens*) skin obtained with three different extraction protocols compared to two commercial fish skin gelatins. We first investigated the gelatin powder composition, and then produced the O/W emulsions at pH 3 by mechanical dispersion followed by an ultrasonication process. Sodium dodecyl sulfate (SDS) profiles for commercial samples indicated that extensive and unspecific hydrolysis of collagen occurred during the production process, whereas gelatin extracted from saithe fish skin showed typical electrophoresis patterns of type I collagen, with the presence of γ- and β-chains. Emulsions obtained with commercial samples presented high physical stability over 7 days, with particle size of ~200 nm. However, emulsions obtained with saithe fish skin presented particle size between 300 and 450 nm. Slight differences were observed in viscosity, with values between ~1 and ~4 mPa·s. Interfacial tension measurements presented values between 13 and 17 mN·m^−1^ with three different regimes for all the systems.

## 1. Introduction

Emulsions are colloidal systems composed of one immiscible liquid dispersed within another liquid in micrometric or nanometric droplets [1]. According to Farjami and Madadlou [2], oil-in-water (O/W) emulsions are widely employed in different areas such as food, pharmaceutical, cosmetic, and paint industries. This is mainly due to their ability to transport or solubilize hydrophobic components in a continuous water phase. Since the emulsification process is entropically unfavorable, this leads to a thermodynamically unstable system [3]. Various physicochemical mechanisms such as coalescence, separation by gravitational force, flocculation, and Ostwald maturation can occur, separating the water and oil phases [1]. Thus, an important challenge is to develop highly stable emulsions with minimal changes in structure during the fabrication process and storage [4]. 

Different processes have been studied to obtain a stable emulsion. Among the different approaches, high energy processes such as rotor–stator emulsification, ultrasound devices, high-pressure homogenizers, and microfluidics are widely employed [5,6]. Ultrasound is considered as a green and sustainable technology employed in the modification of food components and acceleration of certain industrial processes, in particular in liquid systems. Due to a unique cavitation phenomenon called nucleation, droplets grow and break into smaller ones, creating microbubbles. 

Proteins are widely employed in food systems as emulsifiers due to their ability to adsorb to oil–water interfaces and form interfacial films [1,7]. Their surface activity is due to the presence of both hydrophobic and hydrophilic regions in their peptide chains [8]. Proteins move from the bulk to the interface, rearranging themselves to position the hydrophobic group in the oil phase and hydrophilic amino groups in the aqueous phase, reducing interfacial tension and global free energy of the system [8]. 

With changing consumption habits and environmental issues, fish gelatin has become a new alternative in the food industry [9]. Fish gelatin is both (i) a clean label product without religious restrictions and (ii) an ingredient from the valorization of byproducts from marine sources [10]. In the North Atlantic sea, saithe is one of the economically important fish species, with an annual catch of around 300,000 tones [11]. It is used for fillet production, with generation of a large amount of byproducts. However, the species of fish and extraction processes can influence the technofunctional properties of gelatin [10]. Previous studies have validated the fact that fish gelatin can be used as a stabilizer in emulsion systems [12]. However, there are few studies concerning the emulsifying properties of fish skin gelatin. This study aimed to investigate the physical and interfacial properties of O/W emulsion stabilized with gelatin from saithe skin obtained with different extraction processes. Emulsifying properties of three different fish gelatins from saithe (*Pollachius virens*) skin were compared with two fish skin gelatins from the market. Physical stability was characterized using the Turbiscan stability index (TSI), whereas size and charge were characterized by using dynamic light scattering (DLS). Viscosity was analyzed with a rheological test, while interfacial properties were analyzed with the static pendant drop method. Using confocal scanning laser microscopy (CLSM), we analyzed the disposition of proteins on the O/W interface.

## 2. Materials and Methods 

The samples are named and presented in Table 1. Briefly, sample A, gelatin from cold-water fish skin, was purchased from Sigma (Sigma, St. Louis, MO, USA). Sample B, from cold-water fish skin, was purchased from Norland Products Inc. (Cranbury, NJ, USA). Samples C and D were obtained from saithe (*Pollachius virens*) skin, according to Casanova et al. [13]. Sample E was obtained according to [13], with washing of the skin with water. After gelatin extraction, samples were finally spray dried. Fish oil was kindly provided by Lipromar GmbH (Cuxhaven, Germany). 

### 2.1. Physicochemical Characterization 

Water and ash content were determined according to the AOAC (Association of Official Agricultural Chemists) standard methods 930.15 and 942.05, respectively [14]. Total nitrogen content was determined using the Dumas method [15], and a conversion factor of 5.6 was adopted to obtain total protein content. 

### 2.2. Mineral Composition 

The mineral composition was evaluated by inductively coupled plasma mass spectrometry (Thermo iCAPq ICPMS, Thermo Electron, Waltham, MA, USA). A quantity of ~0.3 g was digested with a volume of 5 mL of concentrated nitric acid (SPS Science, Paris, France) in a microwave oven (Multiwave 3000, Anton Paar, Graz, Austria) and then diluted with ultrapure water. External calibration using yttrium as an internal standard was employed to quantify the mineral components. For quality assurance of the results, a certified reference material (DORM-4 fish protein, National Research Council Canada (NRC-CNRC), Food additives, flavours and adulterants) was employed. 

### 2.3. Electrophoretic Study (SDS-PAGE)

Protein profile was determined by polyacrylamide gel electrophoresis, according to Laemmli [16], by using 12% acrylamide (C = 2.6% (*w*/*w*)) slab gels (1.5 mm thick). A quantity of 50 mg of dry sample was extracted in 2 mL 1% sodium dodecyl sulfate (SDS), 100 mM dithiothretiol (DTT), and 60 mM Tris HCl (pH 8.3). After stirring for 1 h at room temperature (T = 20 ± 1 °C), the sample was mixed/homogenized (Polytron PT 1200, Kinematica) for 30 s, then boiled for 2 min and finally incubated for 30 min at room temperature. The sample was homogenized, boiled for 2 min, and centrifuged for 15 min at 20 °C at 20,000× *g*. The supernatant was collected, and an aliquot was diluted with a sample buffer (125 mM Tris HCl (pH 6.8), 2.4% SDS, 50 mM DTT, 10% *v*/*v* glycerol, 0.5 mM EDTA, and bromophenol blue) to load the gel with 10 µL, corresponding to 420 µg protein based on the protein content of the dry sample. The electrophoresis was run at a constant voltage of 100 V for 15 min, followed by 150 V for 1 h. Afterwards, the gel was stained using colloidal Coomassie Brilliant Blue [17]. Mark12^TM^ from Novex was employed for molecular weight markers. 

### 2.4. Emulsion Preparation 

A buffer solution at pH 3 was prepared by dissolving 21 g of citric acid in 200 mL 1 M sodium hydroxide and diluting it to 1000 mL with deionized water. A volume of 80.6 mL of this solution was diluted to 200 mL with 0.1 M of hydrochloric acid. Sodium azide (Sigma, St. Louis, MO, USA) and fish skin gelatin were added to 196 mL of the buffer solution at 0.02% and 4% wt, respectively. The solution was stirred overnight at room temperature (20 ± 1 °C). O/W emulsions (2% fish oil and 98% water) were prepared using Ultra-Turrax (DI 25 Basic, IKA, Staufen, Germany), followed by ultrasonication (SFX550 Model, Branson Ultrasonics Corp., Danbury, CT, USA). The protein suspension was first blended for 10 min (2.5 min × 4) at 8000 rpm by using the Ultra-Turrax, by adding the fish oil drop by drop with a pipette. Then, the emulsion was additionally blended for 10 min (2.5 min × 4) at 20,500 rpm. To avoid temperature increase, the beaker was placed in an ice-water bath. The emulsion was then transferred to a 250 mL beaker, placed in an ice bath, and ultrasonicated for 20 min at 550 Hz in pulsed mode (10 s on and 10 s off) at 100% amplitude.

### 2.5. Emulsion Stability 

A Turbiscan tower (Formulaction, Toulouse, France) was employed to measure the stability of the emulsion over 14 days at 19 °C. A volume of 20 mL of fresh emulsion was put in a vial using a pipette to avoid any significant marks on the glass surface. The vial was placed in the tower, and the physical stability of the emulsion was checked by the scanning program for 7 days as follows: 1 scan every 5 min during the first 6 h; 1 scan every 10 min between 6 and 12 h; 1 scan every 1 h between 12 and 24 h; 1 scan after one day between 1 and 7 days. One scan of the vial from the top to the bottom took 20 s, measuring backscattering (BS) and transmission (T) [18]. Dispersion of the particles, which is related to stability, was measured according to ISO/TR 13097 [19]. BS and T were used for identification of the phenomena related to instability of the emulsion, such as creaming, flocculation, and sedimentation [18]. The instrument reported a stability index, TSI, related to T and BS based on the following equation.
(1)$$TSI\left(t\right)=\frac{1}{N_{h}}\overset{tmax}{\underset{t_{i}=1}{\sum }}\overset{zmax}{\underset{z_{i}=z_{min}}{\sum }}\left|BST\left(t_{i},z_{i}\right)-BST\left(t_{i-1},z_{i}\right)\right|$$
where tmax is the point in time (*t*) when the TSI is measured, $z_{min}$ and $z_{max}$ are the lower and higher selected limits, $N_{h}$ = $z_{max}$ − $z_{min}$/Δh is the high position in the scan, and *BST* is the signal of BS if *T* < 0.2%, otherwise it is *T* [20]. A TSI value higher than 3 indicates that the changes in the matrix of the emulsions became relevant [20]. Measurements were taken three times. 

### 2.6. Dynamic Light Scattering (DLS)

Hydrodynamic diameter (Dh) was determined by DLS on a Zetasizer Nano-S (Malvern Instrument, Worcestershire, UK). Analysis was performed at a scattering angle of 173° and a wavelength of 632.8 nm. Fresh emulsions were firstly diluted 500 times using deionized water. The Dh of particles was calculated by the Stokes–Einstein equation using the diffusion coefficient ($D_{t}$) extracted from (2), as follows:(2)$$Dh=\frac{K_{B}T}{3\pi \eta D_{t}}$$
where $K_{B}$ is Boltzmann’s constant and T is the temperature. 

### 2.7. ζ-Potential Measurements

Zeta-potential (ζ) was determined using a Zetasizer Nano-S (Malvern Instrument, Worcestershire, UK) using capillary cells. Emulsions were diluted 500 times in distilled water. The analyses were performed by applying a voltage of 50 mV. Zeta-potential (ζ) was calculated from the electrophoretic mobility (μ) using Henry’s equation: (3)$$\zeta =\frac{3\;\eta µ}{2\epsilon f\left(kR_{h}\right)}$$
where *η* is viscosity of the buffer (1.033 × 10^−3^ Pa·s^−1^), μ is electrophoretic mobility (V Pa^−1^·s^−1^), and ε is the medium dielectric constant (dimensionless). Debye length (*k*^−1^) measured the thickness of the double electric layer around the molecule (nm). $R_{h}$ (= ¼D_h_/2) is particle hydrodynamic radius (nm), whereas $f\left(kR_{h}\right)$ is Henry’s function. A value of 1.5 was adopted for $f\left(kR_{h}\right)$, which is referred to as the Smoluchowski approximation.

### 2.8. Viscosity Measurements

A controlled-stress HAAKE rheometer (Thermo Fisher, Waltham, MA, USA) equipped with a concentric cylinder (CC 25 Standard) was employed to characterize the flow behavior of fresh emulsions at 19 °C. Steady shear viscosity measurement of emulsions was performed at a shear rate ranging from 25 to 200 s^−1^. Measurements were run 3 times on fresh emulsions. 

### 2.9. Interfacial Properties 

Interfacial tension was measured using the pendant drop method with OCA 25 (DataPhysics Instruments GmbH, Filderstadt, Germany). Before the analysis, the protein solution was sonicated for 20 min in pulsed mode (10 s on and 10 s off). The needle of the syringe was immersed in fish oil and deposed in a glass cuvette. A droplet of 35 μL of protein was formed, and interfacial tension was measured over 420 s and analyzed with SCA20^®^ Software (DataPhysics Instruments GmbH, Filderstadt, Germany). Measurements were repeated 3 times. 

### 2.10. Confocal Scanning Laser Microscopy (CLSM)

A volume of 8 µL of emulsion prepared with Nile red (0.02%) was deposed on a microscope slide with a coverslip (Marienfeld no1.5H) and sealed with nail polish. The sample was imaged at 100× (Nikon CFI plan achromat NA1.45) on a spinning disc confocal microscope (Nikon Ti2). The instrument was equipped with a laser source (405/488/561/640 nm), a confocal spinning disc module (Yokogawa CSU-W1, 50 µm pinholes), a quad-band emission filter (440/521/607/700 nm), and an sCMOS camera (Photometrics Prime95B). Samples were gently placed on a microscope slide and stained with a droplet of aquatic Nile blue 1% wt. 

### 2.11. Statistical Analysis 

The data were tested by ANOVA, and the level of significance was indicated by *p* < 0.05, using Microsoft Excel^®^ software (Redmond, WA, USA). Measurements were taken 3 times for each fresh sample for 3 different preparations. 

## 3. Results

### 3.1. Physicochemical Characterization

Table 2 presents the ash, water, and protein contents of the fish gelatin powders. Samples A and B presented similar values for water. Samples C, D, and E presented lower values with the average value between 5% and 8%. Ash value revealed the presence of three different groups: samples A and B with a value lower than 0.5%, sample E with a value of 1.17%, and samples C and D with a value higher than 24%. Protein content for samples A, B, and E was higher than 80%, whereas samples C and D presented values of ~54% and ~64%, respectively.

### 3.2. Mineral Composition

The analysis of element composition showed two different groups: samples A, B, and E, and samples C and D (Table 3). High levels of Na and Ca were observed for samples C and D, compared to A, B, and E. The order of element levels in gelatin was determined as follows: Sr > Fe > Zn > Cr > Ni > Mn > Cu > Se > Co. A higher quantity of Se and a lower amount of all the other elements was observed for C compared to D. A, B, and E presented lower amounts of microelements compared to samples C and D, except for Fe, Co, and Se. A higher quantity of Cr and Sr was observed for samples C and D. For the toxic elements (Cd, Hg, and Pb), generally low concentrations were found and consequently, no safety issues related to the samples were identified. Low values for Cd were observed in samples B, D, and E (<0.01). Hg content in all the studied samples was lower than 0.01 g/kg dry matter. 

### 3.3. Protein Profile (SDS-PAGE)

Figure 1 illustrates the approximate weight distribution of proteins in samples A, B, C, D, and E. The commercial samples A and B were distinguishable from C, D, and E, obtained from saithe fish skin collagen. Samples A and B presented indistinct bands, indicating that extensive hydrolysis of collagen occurred during the production process. Samples C, D, and E showed electrophoresis patterns typical of type I collagen, with the presence of two α-chains (Mw about 120 kDa), α-1 and α-2, and a *β*-component mainly in C (Mw above 200 kDa) and other high molecular weight aggregates.

### 3.4. Emulsion Stability 

The physical stability of the emulsions was investigated using the Turbiscan tower, and results are presented in Figure 2. It was observed that emulsions stabilized with samples C and D were stable up to 2 days, whereas the emulsion obtained with sample E presented stability for up to 4 days. Emulsions stabilized with gelatins A and B were stable for 7 days with a TSI value of 2.7 and 2.1, respectively. In Figure 3 we present the ΔBS (%) as a function of the height of the cell, from the bottom (0 µm) to the top (40,000 µm). Results are plotted after 0, 7, 13, and 18 h, and 5, 6, and 7 days of emulsion formation. For all the systems, a global trend is observed: at the bottom of the cell (0 µm) ΔBS (%) increased as a function of time, reaching a maximum value of −12%. This behavior means that flocculation of the droplets took place. Between 9000 and 32,000 µm, ΔBS value was close to 0%, whereas when we move towards the top of the cell (up to 40,000 µm), the values increased up to 50%, due to the creaming mechanism. The droplets move and are localized at the top of the cell due to the lower density of the fish oil compared to the water phase.

### 3.5. Hydrodynamic Diameter and ζ-Potential Measurements

Emulsions stabilized by samples A and B presented a stable particle size of ~200 nm for 7 days, as seen in Figure 4a. Similar behavior was observed for sample E, where the size was stable at ~300 nm. On the contrary, emulsions stabilized with samples C and D had a size of 400 and 450 nm at day 0, respectively, with a global tendency to increase up to 500 and 800 nm, respectively, on day 7. ζ-potential for all the samples is presented in Figure 4b. Globally, net charge was 19 ± 7 mV for 7 days. A slight decrease was observed for sample D, where net charge decreased from 22 to 13 mV. 

### 3.6. Flow Behavior

Figure 5 illustrates steady shear viscosity of the prepared emulsions as a function of shear rate. All emulsions exhibited no significant variation in viscosity over the assessed shear rate regime, representing Newtonian flow behavior. Gelatin aqueous solution typically possesses Newtonian behavior at temperatures below its set point [21]. No remarkable difference was observed in viscosity between emulsions B, C, D, and E. They showed viscosity values between 1.2 and 1.8 mPa·s. Viscosity in the emulsion stabilized by sample A (3.5 mPa·s) was higher than in other emulsions. 

### 3.7. Interfacial Tension 

Interfacial tension of the O/W emulsions was investigated with the pendant drop method. As shown in Figure 6a, in the absence of protein in the water phase, interfacial tension is constant at 26 ± 1 mN/m^2^ (black line). When proteins were present in the water phase, a decrease in interfacial tension was observed (*p* < 0.05). Samples A, B, and E reached stable interfacial tension after 200 s with values around 16, 15, and 14 mN/m^2^, respectively. On the contrary, in samples C and D, interface tension was stabilized after 50 s at 13 and 12 mN/m^2^, respectively. In Figure 6b we can recognize the dynamic interfacial tension regimes. Regime II is correlated to the rearrangement of protein structure at the interface with a consequent increase in interfacial contacts [22]. This induces a decrease in interfacial tension. Regime III, for samples A, B, and E, had a similar slope. However, after 50 s, a slight decrease in interfacial tension was observed for samples C and D, probably due to formations of multilayers [23]. 

### 3.8. Confocal Scanning Laser Microscopy (CLSM)

CLSM was used to visualize the gelatin at the O/W interface (Figure 7). Our observations were made principally for the largest droplets in the sample, since the resolution of the microscope did not allow us to clearly distinguish the ring shape corresponding to the O/W interface for all the droplets. Proteins are generally located at the O/W interface. As presented in the middle view, a ring shape made by the proteins was observed; this confirmed that the proteins are only fixed at the O/W interface. However, some more intense points around the droplets were observed, suggesting an aggregation of proteins at the O/W interface and/or small particles of oil that were stuck to the bigger droplet.

## 4. Discussion

### 4.1. Physicochemical Characterization of Gelatins 

We present in this study the physical stability of O/W emulsion stabilized with gelatin from saithe (*Pollachius virens*) skin obtained with three different extraction protocols compared to fish skin gelatin from the market. As presented in Table 1, the high ash content for samples C and D (27.62% and 24.39%, respectively) was related to the presence of Ca and Na (Table 2). This is explained by the absence of a prewashing step of the skin before extraction. As reported by Alfaro et al. [24], the maximum ash content recommended for human consumption is 2.6%. A high amount of Mg was observed for samples C, D, and E. According to Oungbho, Benjakul, Visessanguan, Thiansilakul, and Roytrakul [25], fish gelatins contain low levels of magnesium. Samples C and D presented higher values compared to pork gelatin (0.214 g/kg dry matter [26]). Concerning heavy metals, carp muscle contains 0.016 mg of Cd/kg, 0.11–0.28 mg of Hg/kg, 0.21–0.43 mg of Pb/kg and 0.16–0.17 mg of As/kg [27]. These results are higher than the reported values in this study. This is probably due to the lower tendency of gelatin from saithe skin to accumulate these contaminants. After SDS-PAGE observation, the protein profile for samples A and B was unknown. This was probably due to extensive hydrolysis of collagen. For samples C, D, and E, distinct bands observed at high molecular weight corresponded with the presence of α-chains and β-chains and for sample C, probably also γ-chains. Additional bands at lower molecular weight (<100 kDa), were observed, indicating hydrolysis of elementary chains of collagen or residues of noncollagenous proteins [28]. 

### 4.2. In Bulk Emulsions

O/W emulsions were obtained by mechanical dispersion followed by ultrasonication, as described previously. Microstructural observations of fresh samples using the CLSM approach confirmed the presence of proteins at the O/W interface. In bulk rheology analysis, fresh emulsions presented typical Newtonian behavior. These results can be explained by the lower oil content value (2%). In this case, the viscosity of the system is mainly dominated by the water viscosity phase, as reported by Surh et al. [12]. The higher apparent viscosity of sample A could be attributed to the water–gelatin interaction present in the aqueous phase, electrostatic association between the proteins, and denatured fish gelatin [29,30]. 

### 4.3. Absorption to the Oil–Water Interface

Interfacial properties were investigated by using pendant drop analysis (Figure 6). As expected, all plots showed a decrease in interfacial tension, validating the hypothesis that gelatins can stabilize the oil–water interface. Different kinetics of absorption to the interface were observed for samples A, B, C, D, and E (*p* < 0.05). Gelatins from saithe fish (C, D, and E) and gelatins from the market (A and B) were less able to stabilize interface tension compared to other proteins. Amine et al. [31] showed that pea protein or Na caseinate can decrease interfacial tension around 4 or 2 mN·m^−1^ at pH 7 or 10, respectively. Karefyllakis et al. [32] demonstrated that proteins from sunflower cake show interface tension around 11 mN·m^−1^ after 2000 s. These differences are mainly due to the kinetics and mechanism of protein adsorption from bulk to the interface, protein or peptide size, environmental conditions, and other factors such as the conformation, hydrophobicity, and concentration of the protein [33,34]. At the interface, proteins change their conformation to increase the number of contact points, leading to diminution of interfacial tension. Whey protein hydrolysates decrease interfacial tension less compared to whey protein [22]. Kato and Nakai [35] attributed these differences to the different affinity between the hydrophobic proteins. In our case, the differences were probably mainly due to the unspecific hydrolysis of gelatin (samples A and B). However, the comparison in terms of protein profile with samples C, D, and E remains complicated. Since in drop tensiometry adsorption is mostly related to diffusion from the bulk to the interface, an inverse relation between adsorption rate and peptide size was expected [23,36]. In addition, Vioque et al. [37], in a study of hydrolysis of rapeseed protein isolates, demonstrated that the hydrolysate, with a lower degree of hydrolysis (3.1%), showed the best emulsifying activity and emulsifying stability compared to the nontreated proteins.

### 4.4. Physical Stability of Emulsions

Physical stabilities of emulsions as a function of time were investigated using a Turbiscan tower. As presented in Figure 2, emulsions stabilized with samples C and D were stable up to 2 days, whereas the emulsion obtained with sample E presented stability for up to 4 days. Emulsions stabilized with gelatin A and B presented stability for 7 days. Several hypotheses can be analyzed, firstly by observing the protein contents (%) in the different samples (Table 1). The emulsions stabilized with gelatin C and D (protein content of 71.33% ± 0.34% and 60.34% ± 1.48%, respectively) were less stable over 2 days. On the other hand, the higher protein content for samples A, B, and E (94.24% ± 0.72%; 91.42% ± 0.61% and 96.10% ± 0.34%, respectively) explain the highest stability. However, due to similar protein content in samples A, B, and E, it was observed that commercial samples (A and B) more efficiently stabilized the emulsion than sample E. As discussed, the very mild, unspecific hydrolysis of collagen for samples A and B, compared to sample E, led to different protein patterns (Figure 1). According to Amarowicz [38], enzymatic hydrolysis of proteins can be employed as a powerful tool in the modification of functional properties in food systems. The enzymatic process modifies chain length and hence amino acid composition, affecting the emulsifying properties. Slizyte et al. [39] reported that the most significant factor influencing the emulsifying capacity of protein hydrolysate from cod (*Gadus morhua*) fish byproduct is the amount of water added before hydrolysis. Higher emulsifying capacity was obtained with less water. According to the authors, this can be explained by the plastein reaction that can start at a high concentration of hydrolysate in the system. Results obtained by DLS show two types of evolution of the droplet charge; for the emulsions stabilized with proteins B and D, the trend decreased (from 24 to 12 mV), while that for the emulsions stabilized with samples A, C, and E increased, with values between 17 and 25 mV. The difference in size was observed after 7 days for the O/W emulsion obtained with sample D compared to the other samples. A possible reason is the oxidation of the fish oil leading to a change in charge and size due to the presence of free radicals [40]. During lipid oxidation, surface-active molecules develop, contributing to a destabilization process (i.e., flocculation) of the system [41,42]. 

## 5. Conclusions

Samples A and B (from the market) and E (obtained with prewashing of the skin) presented high protein content (>80%). Samples C and D (without prewashing of the skin) presented lower protein content (up to ~64%) with a higher amount of mineral content (Na, Ca, and Mg). SDS profiles for samples A and B indicated that extensive and unspecific hydrolysis of collagen occurred during the production process, whereas samples C, D, and E showed typical electrophoresis patterns of type I collagen, with the presence of *γ*- and *β*-chains. These differences drove the physical stability of the systems: (i) O/W emulsions obtained with samples A and B presented stability over 7 days, with a particle size of ~200 nm. Gelatin E could stabilize the system over 4 days with a particle size of ~300 nm; (ii) O/W emulsions stabilized with samples C and D were stable for 2 days with a size of 400 and 450 nm at day 0, respectively, with a global tendency to increase up to 500 and 800 nm, respectively, at day 7. Among our different extraction protocols, gelatin E could be employed to encapsulate fish oil in O/W emulsions. However, further investigations will be necessary to understand the role of the proteins at the interface (absorption/desorption mechanism) to tailor a specific enzymatic extraction. 

## Figures and Tables

**Figure 1 foods-09-01718-f001:**
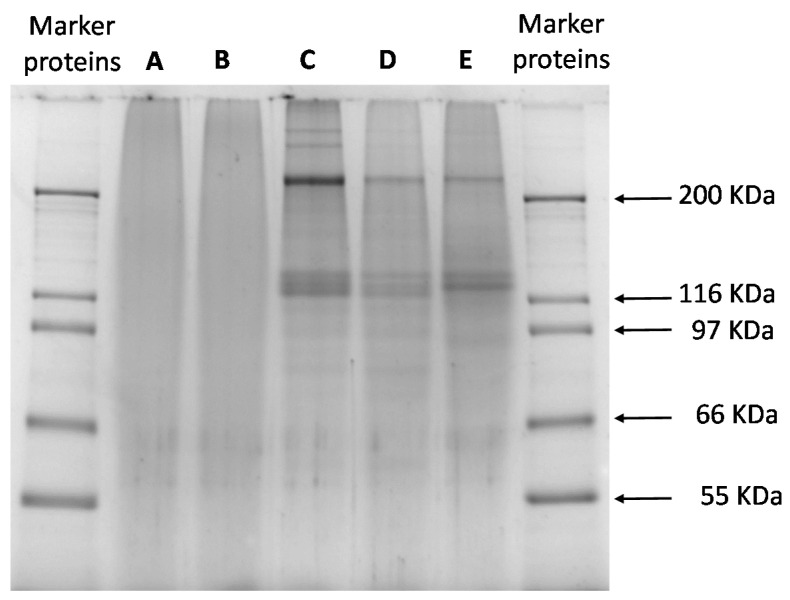
SDS-PAGE for commercial samples A and B from Sigma and Norland, respectively, and C, D, and E from different extraction protocols.

**Figure 2 foods-09-01718-f002:**
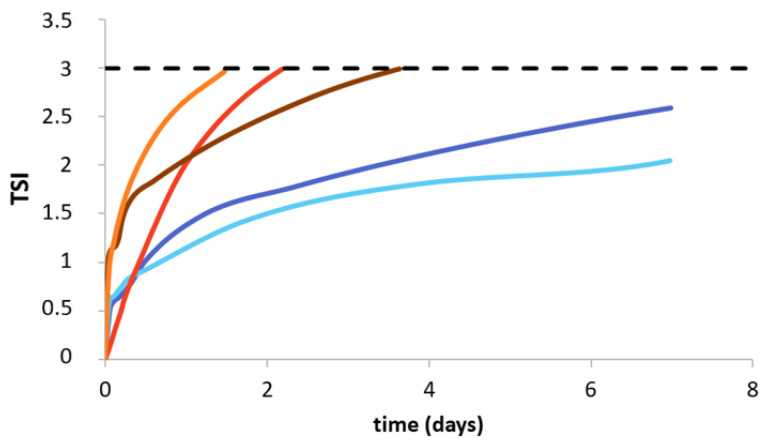
Turbiscan stability index (TSI) as a function of time (days), for Sigma and Norland (A (**-**) and B (**-**)) and C (**-**), D (**-**), and E (**-**).

**Figure 3 foods-09-01718-f003:**
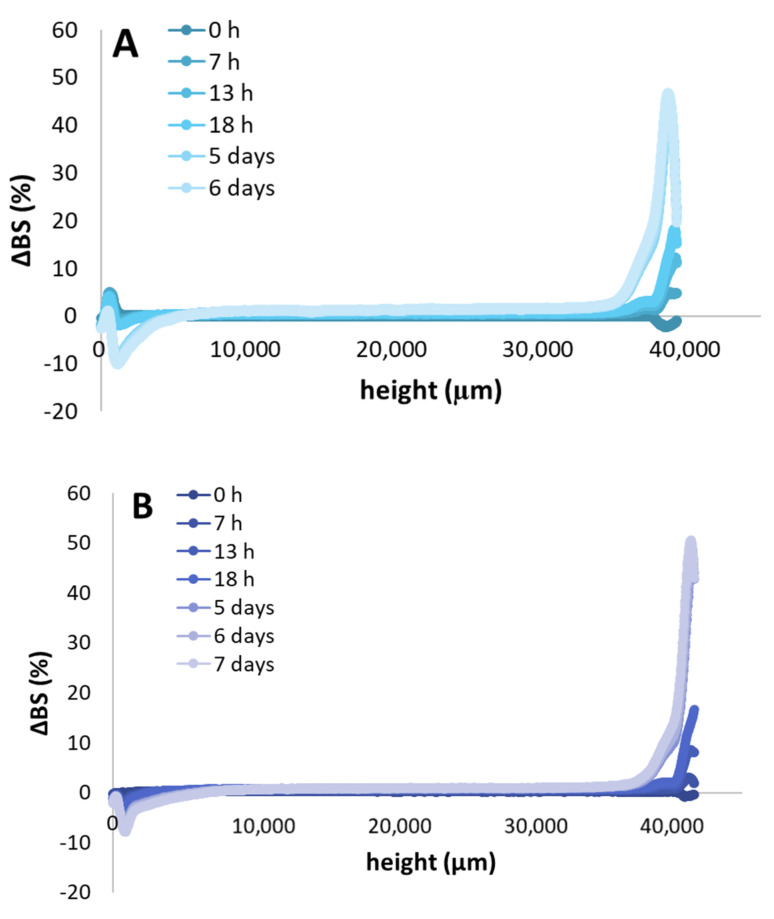
ΔBS (%) as a function of the height (µm) of the cell for Sigma and Norland (**A**,**B**), and from different extraction protocols (**C**–**E**) at time 0 h, 7 h, 13 h, 18 h, 5 days, 6 days and 7 days.

**Figure 4 foods-09-01718-f004:**
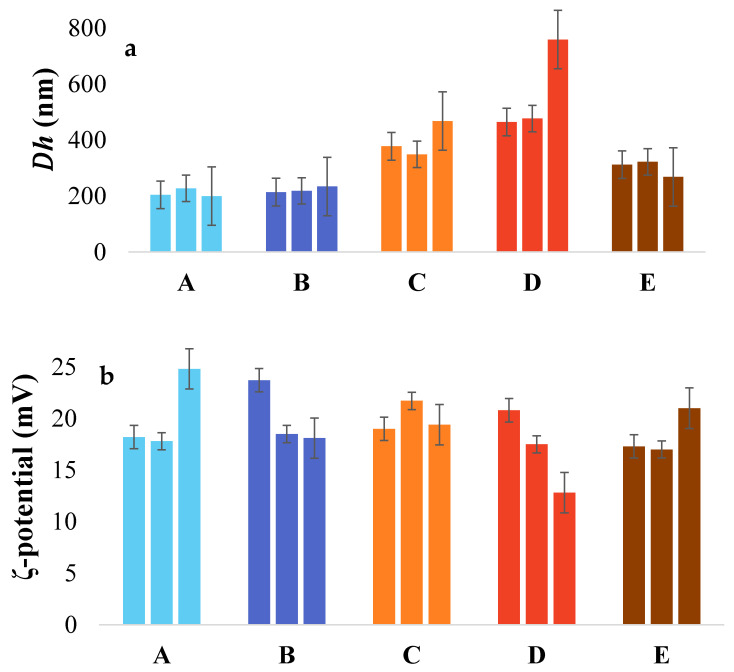
(**a**) Hydrodynamic diameter and (**b**) ζ-potential for commercial samples A and B from Sigma and Norland, respectively, and C, D, and E from different extraction protocols. For each sample we present three histograms correspond to measurements taken at days 0, 4, and 7 (from left to right).

**Figure 5 foods-09-01718-f005:**
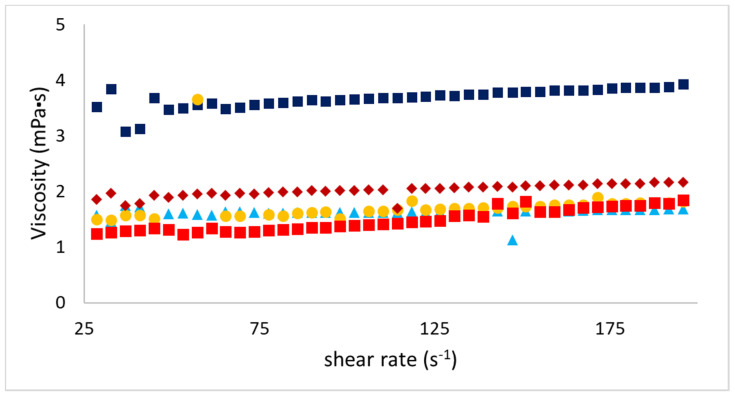
Viscosity (mPa·s) as a function of shear rate (s^−1^) for emulsions A (■), B (▲), C (●), D (■), and E (♦) (T = 19 °C). A and B were obtained from Sigma and Norland, and C, D, and E from different extraction protocols. Error bars are included in the points.

**Figure 6 foods-09-01718-f006:**
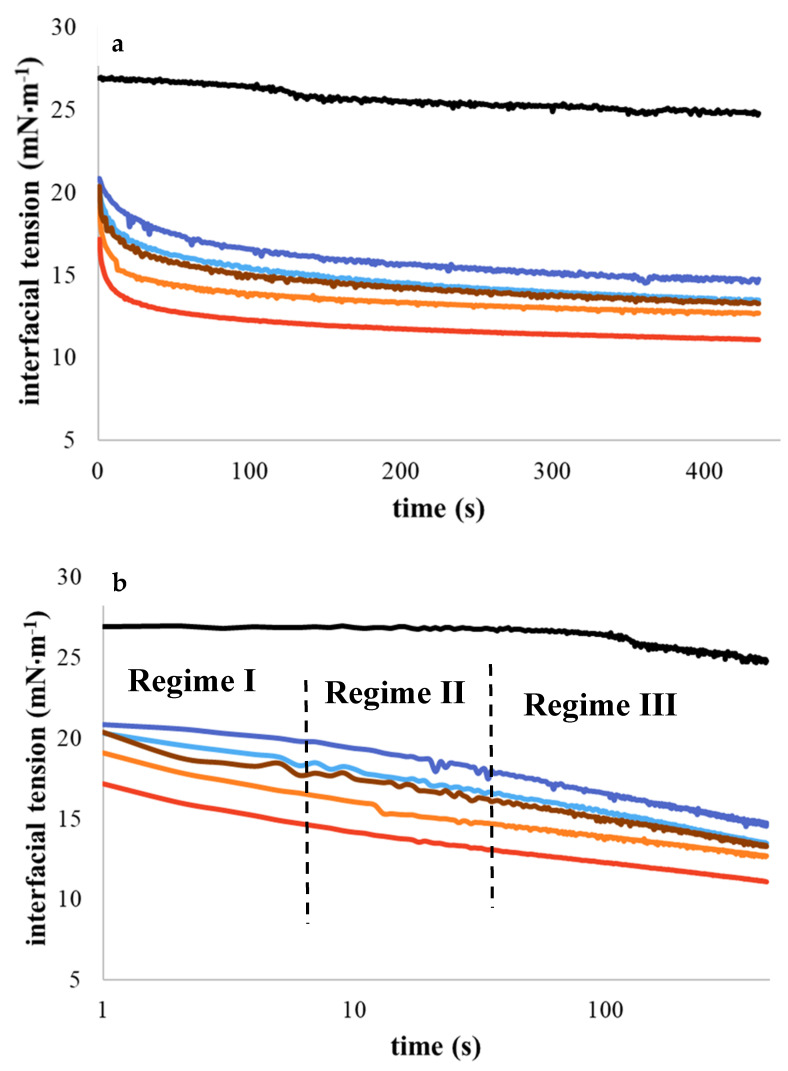
(**a**) Interfacial tension (mN/m^2^) as function of time (s) for emulsions A (-), B (-), C (**-**), D (**-**), and E (**-**). (**b**) Semilogarithmic scale. Black line (**-**) corresponds to the water without proteins. A and B were obtained from Sigma and Norland, and C, D, and E from different extraction protocols.

**Figure 7 foods-09-01718-f007:**
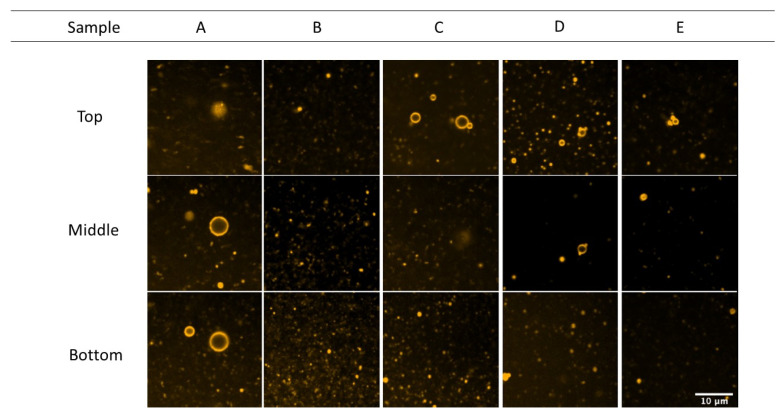
Confocal scanning laser microscopy (CLSM) images at the top, middle, and bottom of the drop emulsions for samples A, B, C, D, and E. This refers to the position in the solution with respect to the two glass surfaces. A and B were obtained from Sigma and Norland, and C, D, and E from different extraction protocols.

**Table 1 foods-09-01718-t001:** Different extraction protocols for gelatin samples C, D, E, and commercial samples A and B.

	Skin Washing	Pre Treatment	Washing	Step 1	Step 2	Step 3	Step 4	Step 5
A	Sigma (commercial sample)							
B	Norland (commercial sample)							
C	No	Neutrase	Water	HCl	80 °C for 5 min	centrifugation 5 min at 2100× *g*	Adjusted pH at 6.5	centrifugation 5 min at 2100× *g*
D	No			Citric acid				
E	Yes							

**Table 2 foods-09-01718-t002:** Water, ash, and protein contents of commercial samples A and B from Sigma and Norland, respectively, and C, D, and E from different extraction protocols.

Sample	Water (%)	Ash (%)	Protein (%)
A	11.73 ± 0.08	0.18 ± 0.03	84.43 ± 0.72
B	11.97 ± 0.01	0.25 ± 0.03	81.91 ± 0.61
C	7.09 ± 0.06	27.62 ± 0.11	63.91 ± 0.34
D	4.81 ± 1.28	24.39 ± 1.06	54.06 ± 1.48
E	7.96 ± 0.13	1.17 ± 0.03	86.10 ± 0.34

**Table 3 foods-09-01718-t003:** Mineral compositions for commercial samples A and B from Sigma and Norland, respectively, and C, D, and E from different extraction protocols.

		A	B	C	D	E
Macroelements (g/kg dry matter)	Na	0.08	0.13	115	72.70	1.43
	Mg	0.01	<0.01	0.88	0.87	0.39
	K	0.16	<0.05	0.75	0.30	1.55
	Ca	0.51	<0.20	2.85	17.70	<0.20
Microelements (mg/kg dry matter)	Cr	0.23	0.11	14.20	18.60	1.60
	Mn	0.01	0.09	1.18	6.38	0.20
	Fe	76.40	1.30	101	127	8.75
	Co	<0.01	<0.01	0.10	0.19	<0.01
	Ni	0.06	<0.01	5.93	9.38	<0.01
	Zn	0.36	<0.01	0.83	97.10	2.83
	Cu	0.45	1.07	2.24	3.79	0.78
	Se	4.57	<0.01	2.35	0.32	<0.01
	Sr	2.16	<0.05	37.20	147	11.1
Toxic elements (mg/kg dry matter)	Cd	0.33	<0.01	0.12	<0.01	<0.01
	Pb	<0.05	<0.05	<0.05	0.14	<0.05
	Hg	<0.01	<0.01	<0.01	<0.01	<0.01

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
