# Peer review of "Physical Stability of Oil-In-Water Emulsion Stabilized by Gelatin from Saithe (Pollachius virens) Skin"

_foods, 2020, doi:10.3390/foods9111718_

Round 1

Reviewer 1 Report

I am never sure why researchers code samples. Please can you replace samples A, B etc. with more useful names liken “commercial fish gelatin 1” or “sigma fish gelatin”, “etc at least help the reader to intuit what the sample is. The way you describe the samples do not even explain how all of them are made – so difference in samples C and D are not only encoded but a mystery too – you cannot expect your reader to read preparation details from other publications. This kind of coding running all the way into the discussions and conclusions means that the paper never tells the reader what is being compared.  It really makes the paper into a nonsense.

You talk about “rheology” and “viscosity” as though they are always measured the same way.  How did you measure it? How were samples prepared for viscosity measurement? It is not actually clear if the viscosity is for gelatin solutions or for emulsions, please can you clarify this. Did you have similar dry matter present to make similar concentrations of gelatin solutions – I cannot find this information.  Figure 5 has some “stray readings” (especially sample c) how many replicates involved in this work? Can you provide error bars to show variation? In addition to stray readings, are there missing readings?

Please can you rename “moisture” as “water”.

How was protein determined? Is it based on N content and if so what conversion factor is use? In table 1, either a remove the word “content” from the protein column – be consistent.

Are you really measuring high levels of Strontium in your samples?

Figure 2 & 3 – the legend colours do not match the curve colours.

Please check use of symbols e.g. k in kilo prefix is always lowercase.

Author Response

I am never sure why researchers code samples. Please can you replace samples A, B etc. with more useful names liken “commercial fish gelatin 1” or “sigma fish gelatin”, “etc at least help the reader to intuit what the sample is. The way you describe the samples do not even explain how all of them are made – so difference in samples C and D are not only encoded but a mystery too – you cannot expect your reader to read preparation details from other publications. This kind of coding running all the way into the discussions and conclusions means that the paper never tells the reader what is being compared.  It really makes the paper into a nonsense.

Dear Reviewer, we understand your point of view. However, in order to simplify the text, tables and figures, we are strongly encouraged from our Community, to adopt this kind of listing. Sample are introduced in the section “Materials and Methods”, in the new lines 73-78. To obtain more detailed information, you can read the reference 13, published by our group (and myself as first author). Then, all around the Manuscript we chose to call them as A, B, C, D and E.  

You talk about “rheology” and “viscosity” as though they are always measured the same way.  How did you measure it? How were samples prepared for viscosity measurement? It is not actually clear if the viscosity is for gelatin solutions or for emulsions, please can you clarify this.

We applied the rheology analysis to measure the viscosity of our preparation, as explained in the new lines 160-163. We clarify this in the new lines 157-160, the viscosity is measured on the fresh emulsions.

Did you have similar dry matter present to make similar concentrations of gelatin solutions – I cannot find this information. 

This information is present in Table 1. We chose to work with the same quantity of protein, even for the sample C and D, where the dry matter content is higher compared to A, B and E.

Figure 5 has some “stray readings” (especially sample c) how many replicates involved in this work? Can you provide error bars to show variation? In addition to stray readings, are there missing readings?

Analysis were repeated three times and the error bars are included in the points. We add this information in the new lines 271-272.

Please can you rename “moisture” as “water”.

We replace water instead of moisture.

How was protein determined? Is it based on N content and if so what conversion factor is use? In table 1, either a remove the word “content” from the protein column – be consistent.

We add this information in the section “Material and Methods”, in the new lines 80-82. We eliminate the word “content” from the protein column, in Table 1.

Are you really measuring high levels of Strontium in your samples?

Yes.

Figure 2 & 3 – the legend colours do not match the curve colours.

As requested, we change the color legend.

Please check use of symbols e.g. k in kilo prefix is always lowercase.

We check all the symbols.

Reviewer 2 Report

The manuscript of Henriet and coworkers deals with the physical stability of oil-in-water emulsion stabilized by gelatin from saithe (Pollachius virens) skin. The manuscript is globally well written, the conclusion are sound and th experimental section is well described. Indeed, in the introduction, I propose to describe the Pickering emulsions obtained with protein (or perhaps with cyclodextrins) and the difference with your system (in the conclusion).

Author Response

The manuscript of Henriet and coworkers deals with the physical stability of oil-in-water emulsion stabilized by gelatin from saithe (Pollachius virens) skin. The manuscript is globally well written, the conclusion are sound and the experimental section is well described. Indeed, in the introduction, I propose to describe the Pickering emulsions obtained with protein (or perhaps with cyclodextrins) and the difference with your system (in the conclusion).

We understand the point of view of the reviewer. However, the pickering emulsions obtained with protein or with cyclodextrins, is not the focus of this Manuscript. In this context, we need to describe all the possible pickering stabilizing agents (protein-based) revised in the literature. The same will be for cyclodextrins. In our case, we work with soluble protein i.e. gelatin (and not other molecules), stabilizing O/W emulsions by a single layer around the droplet (and not with particle). In conclusion, is not possible to compare these systems.

Reviewer 3 Report

The authors present a study focused on the physical stability of oil-in-water emulsion using gelatin from saithe (Pollachius virens) skin, prepared by different extraction methods. The information presented is of interest to the food industry due to the valorization of waste materials and the use of fish gelatin, which is important for specific markets.

The paper needs editing for English, some areas more than others (abstract, introduction and M&M need more work). In general, the paper must improve the way the tables and figures are presented, with much more complete and better descriptions. Statistical differences are not shown in any Tables or Figures. Materials and Methods section is also lacking key information.  The Results section is not strong enough to stand alone, it should be combined with the Discussion.

Abstract

Include main quantitative results; lacks main findings, significant improvements needed

Introduction

The introduction is missing key references related to fish gelatin.  Authors are encouraged to add additional references to improve this section.

Lines 62-63 “With the actual trend, the change of habits consumption and environmental issues, fish gelatin has become a new alternative in the food industry [9].” This sentence needs further clarification and many additional references.

Materials and Methods

Lines 79-80: Sample A and Sample B – add names and more description.   “purchased by from…

Lines 81-82: add a short description of how the skin was obtained and skin particle size used for extraction.  How are samples C and D different from each other? Why was E prepared in a different way? Need to add a justification as the entire study is based on the differences in extraction methods. Suggest to add a diagram explaining the difference in the extraction methods, that will help with the discussion.

Lines 86-89: it is not clear what is the final fish gelatin sample, need to add better explanation.

Lines 91-92: What sample is analyzed?  Is the final sample dried? How much sample is used in each analysis? Not clear.

Line 94: How much sample was used for the analysis?

Line 118: need a reference for emulsion preparation.

Lines 186: “Measurements were realized 3 times on fresh samples” Does this mean that fresh emulsions were prepared 3 times for each analysis? Or 1 fresh emulsion was prepared and analyzed 3 times?

Results

Lines 189-194: Information presented is repetition of what is presented in Table 1, eliminate repetition.

Table 1: Need to add a description of the samples instead of referring to them as A…E. Indicate what the data presented is, mean and standard deviation? Of 3 measurements?

Table 2: Change the title to “Metal (or mineral) concentration…” instead of “Elemental composition”. Also, same comments as Table 1, add standard deviation to means.

Lines 215-216, and Figure 1: correct KDa to kDa

Figure 2: add more information to the title. “Stability Index (TSI) as a function of time at X°C for emulsions prepared with gelatin samples…..”.  If measurements were replicated, need to show the variation due to the replicates.

Figure 3: Need a more complete title/description of what the figure presents.

Figure 4: Better and more complete description is needed. Mean and standard deviation? What does “Table 0. -7 respectively” mean?

Figure 5: Again, show the variation due to the replication.  Title not complete (viscosity of emulsions, not samples!)

Figure 6: same comments as Fig 5

Figure 7: spell out CLSM, better description needed.

Discussion

Seems well organized and should be presented as Results and Discussion combined – the separate results section as written is quite weak, and does not work for this paper.

Author Response

The authors present a study focused on the physical stability of oil-in-water emulsion using gelatin from saithe (Pollachius virens) skin, prepared by different extraction methods. The information presented is of interest to the food industry due to the valorization of waste materials and the use of fish gelatin, which is important for specific markets.

The paper needs editing for English, some areas more than others (abstract, introduction and M&M need more work). In general, the paper must improve the way the tables and figures are presented, with much more complete and better descriptions. Statistical differences are not shown in any Tables or Figures. Materials and Methods section is also lacking key information.  The Results section is not strong enough to stand alone, it should be combined with the Discussion.

Abstract

Include main quantitative results; lacks main findings, significant improvements needed.

We agree with the reviewer but we prefer just to add a final sentence concerning the physical stability, which is the topic of this study.  

Introduction

The introduction is missing key references related to fish gelatin.  Authors are encouraged to add additional references to improve this section.

We agree with the reviewer and we improve this section. Please, find the new version in the new lines 34-71.

Lines 62-63 “With the actual trend, the change of habits consumption and environmental issues, fish gelatin has become a new alternative in the food industry [9].” This sentence needs further clarification and many additional references.

We agree with the reviewer and we improve this section.

Materials and Methods

Lines 79-80: Sample A and Sample B – add names and more description.   “purchased by from…

We agree with the reviewer and we improve this section. However, more information are listed in our previous Manuscript, as reported by the reference 13.

Lines 81-82: add a short description of how the skin was obtained and skin particle size used for extraction.  How are samples C and D different from each other? Why was E prepared in a different way? Need to add a justification as the entire study is based on the differences in extraction methods. Suggest to add a diagram explaining the difference in the extraction methods, that will help with the discussion.

We add more information and, as explained in the previous comment, more information are listed in our previous Manuscript, as reported by the reference 13. We think that the diagram is not necessary since this protocol extraction is quite common and a lot Manuscript were published before with this approach.

Lines 86-89: it is not clear what is the final fish gelatin sample, need to add better explanation.

The gelatin sample is the powder, obtained after spray-drying, as reported in the new line 78.  

Lines 91-92: What sample is analyzed?  Is the final sample dried? How much sample is used in each analysis? Not clear.

The physico-chemical characterization was realized based on AOAC standard methods 930.15 and 942.05, respectively, as explained in the new lines 80-82. This is an international standard method of analysis.

Line 94: How much sample was used for the analysis?

Please, check the AOAC standard methods 930.15 and 942.05.

Line 118: need a reference for emulsion preparation.

This is an home-made protocol. We create this in our lab. Is for this reason that we don’t add a reference.

Lines 186: “Measurements were realized 3 times on fresh samples” Does this mean that fresh emulsions were prepared 3 times for each analysis? Or 1 fresh emulsion was prepared and analyzed 3 times?

3 times on fresh samples” means that for each preparation we measure three times.

Results

Lines 189-194: Information presented is repetition of what is presented in Table 1, eliminate repetition.

We agree with the reviewer and we change the text, in the new lines 181-186.

Table 1: Need to add a description of the samples instead of referring to them as A…E. Indicate what the data presented is, mean and standard deviation? Of 3 measurements?

The sample were described in the section “Material and Methods”. However, in order to simplify the text, tables and figures, we are strongly encouraged from our Community, to adopt this kind of listing. Sample are introduced in the section “Materials and Methods”, in the new lines 73-78.  

Table 2: Change the title to “Metal (or mineral) concentration…” instead of “Elemental composition”. Also, same comments as Table 1, add standard deviation to means.

We agree with the reviewer and we change the title of this section. We don’t need the error bars because the values were in good agreement with the certified values (recoveries in the range of 83-116% of certified values for all elements).

Lines 215-216, and Figure 1: correct KDa to kDa

We agree with the reviewer and we change in Figure 1 and in the new lines 219-220.

Figure 2: add more information to the title. “Stability Index (TSI) as a function of time at X°C for emulsions prepared with gelatin samples…..”.  If measurements were replicated, need to show the variation due to the replicates.

We agree with the reviewer and we change in Figure 2.

Figure 3: Need a more complete title/description of what the figure presents.

We agree with the reviewer but more explications are present in the new lines 245-246. 

Figure 4: Better and more complete description is needed. Mean and standard deviation? What does “Table 0. -7 respectively” mean?

We agree with the reviewer and we add more information in the new line 258-260. 

Figure 5: Again, show the variation due to the replication.  Title not complete (viscosity of emulsions, not samples!)

We agree with the reviewer and we add more information in the new line 271-272. Error bars are included in the points.

Figure 6: same comments as Fig 5

We agree with the reviewer and we add more information in the new line 287-288. All the curves overlaps and for this reason we present just one, without any standard deviation. 

Figure 7: spell out CLSM, better description needed.

We agree with the reviewer and we provide the information. 

Discussion

Seems well organized and should be presented as Results and Discussion combined – the separate results section as written is quite weak, and does not work for this paper.

We don’t agree with the reviewer. We think that, for this Manuscript, the best way to present our investigation is to divide the results and the discussion. We present first the results obtained and then we analyze them in three different sections and we compare with the previous work published in the literature. 

Round 2

Reviewer 1 Report

In my first review of this manuscript I asked you to replace samples A, B etc. with more useful names liken “commercial fish gelatin 1”.  It might seem simpler for you to continue like this, but it is not simpler for the reader of your work. Moreover, to expect the reader to research your other papers so as to understand your present submission is arrogant.  I would advise all authors not pick fights with reviewers – if you want to publish then consider their suggestions with a bit more grace – who knows other people might find it easer to follow and understand your work as well as the reviewer!!

In my first review of this manuscript I asked about “rheology” and “viscosity”   in your response you directed me to lines 160-163 but these are about interfacial properties - please can you respond to questions with appropriate line numbers.

What should we believe from table 1 – samples A, B and E appear to have well over 100% I know these are averages, but something is not right!.  Perhaps the answer relates to the conversion factor of 6.25 which you use to calculate the protein. I strongly suggest that you read this article https://doi.org/10.1080/10408390701279749 and reconsider the conversion factor that is used in this work!

If in Table 5 the error bars are included, then I cannot see them. Is it possible that  the symbols for each point are too large and obscure the error bars

Author Response

Dear Reviewer,

To facilitate the lecture, all the changes in the Manuscript are in red.

Best,

Federico Casanova and collaborators.

In my first review of this manuscript I asked you to replace samples A, B etc. with more useful names liken “commercial fish gelatin 1”.  It might seem simpler for you to continue like this, but it is not simpler for the reader of your work. Moreover, to expect the reader to research your other papers so as to understand your present submission is arrogant.  I would advise all authors not pick fights with reviewers – if you want to publish then consider their suggestions with a bit more grace – who knows other people might find it easer to follow and understand your work as well as the reviewer!!

  • Dear Reviewer, It was not our intention to be arrogant with the reader. We agree with you that is easier to understand the Manuscript with other user names. We intend to describe the sample in the section “Material and Methods” and to keep the name as A, B, C, D and E for a simple reason of simplicity, in the Manuscript and especially in the Figure. From our point of view, more text to explain the samples (instead of A, B, C, D and E or 1, 2, 3, 4 and 5 or a, b…) means increasing in size of the Tables and a reduction in the Images space. However, we add Table 1, to better explain the difference between each sample extractions and we add more description in the Tables and Figures captions.

In my first review of this manuscript, I asked about “rheology” and “viscosity”   in your response you directed me to lines 160-163 but these are about interfacial properties - please can you respond to questions with appropriate line numbers.

  • We agree with the reviewer and it was our mistake. The new lines are 169-172.   

What should we believe from table 1 – samples A, B and E appear to have well over 100% I know these are averages, but something is not right!.  Perhaps the answer relates to the conversion factor of 6.25 which you use to calculate the protein. I strongly suggest that you read this article https://doi.org/10.1080/10408390701279749 and reconsider the conversion factor that is used in this work!

  • We agree with the reviewer and we adopt 5.6 as a protein conversion factor, as suggested in the peer-reviewed Manuscript (https://doi.org/10.1080/10408390701279749). We change the values in line 94 and in Table 2.

If in Table 5 the error bars are included, then I cannot see them. Is it possible that  the symbols for each point are too large and obscure the error bars

  • We understand the reviewer. In Figure 5 we repeated the analysis 3 times on 3 different preparations. The analysis revealed very similar results. This low viscosity and high levels of repeatability is due to the low oil content in the O/W emulsions.  In conclusion, the error bars are small and included in the point. From a graphical point of view are a very small point inside the symbols. We chose this size and color of symbols since, from a graphical point of view, is easier for the readers to understand the Figure.

Reviewer 3 Report

The paper was improved based on corrections.

  1. The paper still needs editing for English.

  1. Abstract: Include main quantitative results; lacks main findings, significant improvements needed. Authors did not improve the abstract.

  1. Materials and Methods: Why was E prepared in a different way? Need to add a justification as the entire study is based on the differences in extraction methods. I still believe that adding a diagram explaining the difference in the extraction methods will help to visualize the differences, as the samples are only listed as letters throughout the paper (including tables and figures).

  1. Statistical analysis: “Measurements were realized 3 times on fresh samples”. Authors clarified that the emulsions were prepared one time and analyzed 3 times. This is a key flaw of the research, the authors are only measuring the error/variability of the testing, not the variability due to the preparation of the emulsion using different gelatins.

  1. Results: Information presented in lines 181 to 186 is repetition of what is presented in Table 1, eliminate repetition.

Table 2: Change the title to “Metal (or mineral) concentration…” instead of “Elemental composition”.

Figure 2: add more information to the title. “Stability Index (TSI) as a function of time at X°C for emulsions prepared with gelatin samples….”

Figure 3: Need a more complete title/description of what the figure presents.

Figure 4: Better and more complete description is needed. Mean and standard deviation? What does “Table 0. -7 respectively” mean? That last sentence makes no sense!

Author Response

Dear Reviewer,

To facilitate the lecture, all the changes in the Manuscript are in red.

Best,

Federico Casanova and collaborators.

  1. The paper still needs editing for English.
  • We agree with the reviewer and we provide the English revision of the Manuscript via mdpi website.   

  1. Abstract: Include main quantitative results; lacks main findings, significant improvements needed. Authors did not improve the abstract. 
  • We agree with the reviewer and we re-write the abstract including the new finding in the new lines 21-32.  

  1. Materials and Methods: Why was E prepared in a different way? Need to add a justification as the entire study is based on the differences in extraction methods. I still believe that adding a diagram explaining the difference in the extraction methods will help to visualize the differences, as the samples are only listed as letters throughout the paper (including tables and figures).
  • We agree with the reviewer and we re-write the paragraph in the new lines 79-84. We add Table 1 to better explain the different protocol extractions. Sample E we “pre-wash” the skin water. This is the main difference between sample C and D respect to E. This “pre-washing” of the skin induces a decrease in the mineral composition, as presented in Table 3, in the case of the elements Na and Ca. This is one of the main objectives of this investigation.    

  1. Statistical analysis: “Measurements were realized 3 times on fresh samples”. Authors clarified that the emulsions were prepared one time and analyzed 3 times. This is a key flaw of the research, the authors are only measuring the error/variability of the testing, not the variability due to the preparation of the emulsion using different gelatins.
  • We understand the reviewer but it was our mistake in the description of the statistical analysis. We clarify this point in the new lines 189-190.

  1. Results: Information presented in lines 181 to 186 is repetition of what is presented in Table 1, eliminate repetition.
  • We agree with the reviewer and we improve this paragraph in the new lines 193-198.

Table 2: Change the title to “Metal (or mineral) concentration…” instead of “Elemental composition”.

  • We change with “Mineral composition” instead of “Elemental composition”.

Figure 2: add more information to the title. “Stability Index (TSI) as a function of time at X°C for emulsions prepared with gelatin samples….”

  • We agree with the reviewer and we corrected, in the new lines 248-249.

Figure 3: Need a more complete title/description of what the figure presents.

  • We agree with the reviewer. We made the changes in the new lines 255-256.  

Figure 4: Better and more complete description is needed. Mean and standard deviation? What does “Table 0. -7 respectively” mean? That last sentence makes no sense!

  • We agree with the reviewer. We made the changes in the new lines 267-269.  
